# Antarctic Thraustochytrids as Sources of Carotenoids and High-Value Fatty Acids

**DOI:** 10.3390/md19070386

**Published:** 2021-07-06

**Authors:** Allison Leyton, Liset Flores, Carolina Shene, Yusuf Chisti, Giovanni Larama, Juan A. Asenjo, Roberto E. Armenta

**Affiliations:** 1Center for Biotechnology and Bioengineering (CeBiB), Center of Food Biotechnology and Bioseparations, BIOREN and Department of Chemical Engineering, Universidad de La Frontera, Francisco Salazar 01145, Temuco 4780000, Chile; allison.leyton@ufrontera.cl (A.L.); liset.flores@ufrontera.cl (L.F.); 2School of Engineering, Massey University, Private Bag 11 222, Palmerston North 4442, New Zealand; Y.Chisti@massey.ac.nz; 3Centro de Modelación y Computación Científica, Universidad de La Frontera, Av. Francisco Salazar 01145, Temuco 4780000, Chile; giovanni.larama@ufrontera.cl; 4Centre for Biotechnology and Bioengineering (CeBiB), Department of Chemical Engineering and Biotechnology, Universidad de Chile, Beauchef 851, Santiago 8370459, Chile; juasenjo@ing.uchile.cl; 5Mara Renewables Corporation, 101A Research Drive, Dartmouth, NS B2Y 4T6, Canada; rarmenta@maracorp.ca; 6Department of Process Engineering and Applied Science, Dalhousie University, Halifax, NS B3H 4R2, Canada

**Keywords:** *Thraustochytrium*, antarctic thraustochytrids, microbial carotenoids, canthaxantin, docosahexaenoic acid, eicosapentaenoic acid

## Abstract

Eicosapentaenoic acid (EPA), docosahexaenoic acid (DHA), and carotenoids are needed as human dietary supplements and are essential components in commercial feeds for the production of aquacultured seafood. Microorganisms such as thraustochytrids are potential natural sources of these compounds. This research reports on the lipid and carotenoid production capacity of thraustochytrids that were isolated from coastal waters of Antarctica. Of the 22 isolates, 21 produced lipids containing EPA+DHA, and the amount of these fatty acids exceeded 20% of the total fatty acids in 12 isolates. Ten isolates were shown to produce carotenoids (27.4–63.9 μg/g dry biomass). The isolate RT2316-16, identified as *Thraustochytrium* sp., was the best producer of biomass (7.2 g/L in five days) rich in carotenoids (63.9 μg/g) and, therefore, became the focus of this investigation. The main carotenoids in RT2316-16 were β-carotene and canthaxanthin. The content of EPA+DHA in the total lipids (34 ± 3% *w/w* in dry biomass) depended on the stage of growth of RT2316-16. Lipid and carotenoid content of the biomass and its concentration could be enhanced by modifying the composition of the culture medium. The estimated genome size of RT2316-16 was 44 Mb. Of the 5656 genes predicted from the genome, 4559 were annotated. These included genes of most of the enzymes in the elongation and desaturation pathway of synthesis of ω-3 polyunsaturated fatty acids. Carotenoid precursors in RT2316-16 were synthesized through the mevalonate pathway. A β-carotene synthase gene, with a different domain organization compared to the gene in other thraustochytrids, explained the carotenoid profile of RT2316-16.

## 1. Introduction

Long chain ω-3 polyunsaturated fatty acids (PUFA), eicosapentaenoic acids (EPA, C20:5n−3) and docosahexaenoic acids (DHA, C22:6n−3), and carotenoids play important roles in human metabolism. DHA is involved in neural and retinal development, aging, memory formation, synaptic membrane function, photoreceptor biogenesis and function, and vision. It is concentrated in the central nervous system and the retina and is esterified to phospholipids, the major component of cell membranes [1]. DHA has potent anti-inflammatory properties and other beneficial effects on chronic diseases such as coronary heart disease, asthma, rheumatoid arthritis, and osteoporosis [2]. Although some physiological effects of EPA and DHA overlap, they differ in their enzymatic conversion to lipid mediators, the structural characteristics/properties of the phospholipids in which they incorporate, and in their metabolic roles [3,4]. Similarly, carotenoids are commercially important fat-soluble natural food colorants with reported health benefits. Biological activities of carotenoids include provitamin A activity (essential for the promotion of growth, embryonal development, and visual function), cellular signaling, and antioxidant activity [5]. Some marine yeasts [6], microalgae [7], and protists such as thraustochytrids are natural producers of EPA, DHA, and carotenoids.

Thraustochytrids are heterotrophic single-celled eukaryotes found commonly in marine ecosystems, and many of them have been investigated as potential sustainable sources of DHA to replace fish oil derived DHA from wild-caught marine fish. Much interest has focused on the thraustochytrid genus *Schizochytrium*, as species within this genus are particularly rich in lipids (30–50% of dry biomass), having a high proportion of DHA [8]. Other thraustochytrid genera, especially *Thraustochytrium* spp., produce lipids with a more complex fatty acid profile that includes DHA, EPA, and arachidonic acid (ARA, C20:4n−6) [9,10,11]. These long-chain PUFA occur in both neutral lipids and the polar lipids of thraustochytrids [12].

Some thraustochytrids also produce carotenoids [13,14,15,16,17]. The amount and composition of carotenoids depend on the producing species and their growth conditions (media composition and temperature). Such growth conditions also influence the PUFA profile of the total lipids [18,19,20]. Carotenoids reported in thraustochytrids include ketocarotenoids such as astaxanthin, canthaxanthin, phoenicoxanthin and echinenone, and non-ketocarotenoids such as β-carotene and zeaxanthin [21]. Although the role of carotenoids in photosynthetic organisms is well known, their function in non-photosynthetic organisms such as thraustochytrids is unclear. Thraustochytrids in their habitats often experience extreme and fluctuating conditions. In such environments, the antioxidative capacity of carotenoids may contribute to protecting the cells against various environmental stresses. Indeed, carotenoids have been shown to confer a protective effect against freeze-thaw stress and solar radiation in certain Antarctic heterotrophic bacteria [22].

A previous study of a different Antarctic thraustochytrid (*Oblongichytrium* sp.) had shown a low culture temperature (5 °C) to promote accumulation of EPA and DHA in total lipids [19], but this microorganism did not accumulate carotenoids. Therefore, a subsequent search was carried out to screen for novel thraustochytrids which are able to produce carotenoids in addition to EPA and DHA. The focus of the search was the cold marine coastal environment of Antarctica, which naturally provides a suitable selection pressure for accumulation of long-chain PUFA.

This research reports on the lipid and carotenoid production capacity of thraustochytrids that were isolated from coastal waters of Antarctica. The genome of the most prolific lipid producing strain, with the capacity to produce lipids rich in long chain ω-3 PUFA and carotenoids, was sequenced. Furthermore, the selected strain’s metabolic pathways used in the synthesis of these valuable compounds were elucidated in attempts to identify strategies for a future lipid yield improvement and for monitoring the expression of critical genes under different culture conditions.

## 2. Results

### 2.1. Lipid and Carotenoid Accumulation

Twenty-two strains isolated from seawater samples were grouped according to their ability to accumulate carotenoids. Identification results for the 12 pale isolates (carotenoid-lacking white colonies on agar plates) are shown in Table 1. These isolates were closely related to *Thraustochytrium* sp. (1 strain), *Aurantiochytrium* sp. (1 strain), and *Oblongichytrium* sp. (10 strains). The ten isolates that accumulated carotenoids were closely related to *Thraustochytrium* sp. (9 strains) and *Aurantiochytrium* sp. (1 strain). (Table 2). Some morphological information (colony size and texture, color of the colony, presence of motile zoospore) for the different isolates is shown in Appendix A. Light microscope images (40×) of some of the isolated thraustochytrid strains are shown in Appendix A.

In the phylogenetic tree, constructed using the 18S rRNA gene sequences Appendix A, nine of the carotenogenic isolates were grouped in one branch. Nine of the non-carotenogenic isolates clustered in a second branch that included *Schizochytrium* sp. ATCC 20888, *Ulkenia profunda* BUTRBG 111, and *Aurantiochytrium* CCAP-406 2/3, proposed as a member of a new thraustochytrid genus *Hondaea* [23].

The total content of carotenoids in the biomass of the pigmented isolates ranged from 27.4 ± 5.3 (RT2316-38) to 63.9 ± 3.2 (RT2316-16) μg/g (Table 2). Under identical culture conditions, two of the carotenogenic isolates (RT2316-44, RT2316-49; Table 2) attained a biomass concentration of 8.3 g/L, or 2.1-fold more than the highest concentration recorded among the non-carotenogenic isolates (RT2316-28; Table 1).

### 2.2. Thraustochytrium sp. RT2316-16

The high biomass concentration and high levels of carotenoids and lipids in the biomass of RT2316-16 (Table 2) made the isolate *Thraustochytrium* sp. a potential candidate for producing carotenoids and lipids. For this isolate, the profiles of its biomass concentration and metabolites (carotenoids, lipids) production during a 10-day batch culture are shown in Figure 1.

Around 97% of the initial glucose was consumed by day five and, at this point, biomass concentration had reached 8.8 ± 0.1 g/L with a relatively high lipid content (34 ± 3% *w/w*) in the biomass (Figure 1). During the same period, the residual amino acid concentration declined from 6.0 ± 0.2 g/L to nearly 0.8 ± 0.2 g/L coinciding with cessation of growth (Figure 1a). The total lipid content in the biomass of RT2316-16 increased 1.7-fold during growth (Figure 1b). A significant decrease (25%) in the total lipid content was observed once glucose was exhausted (Figure 1b). The total carotenoid in the biomass increased 2.9-fold during growth (Figure 1b). A qualitative thin layer chromatography (TLC) analysis of the carotenoids showed the main carotenoids to be β-carotene and canthaxanthin Appendix A. Fatty acid composition of the total lipids in the biomass of RT2316-16 varied during the 10-day culture period (Figure 2).

The main saturated fatty acids in the total lipids were myristic acid (C14:0; 3.4–9.4% of the total fatty acids) and palmitic acid (C16:0; 30.9–49.3%). The total lipid fraction contained the unsaturated fatty acids pamitoleic acid (C16:1n−7; 2.7–8.0 %), linolelaidic acid (C18:2n−6t; 0–23.7%) and γ-linolenic acid (C18:3n−6; 3.7–13.3%). The percentage of EPA + DHA (14.0–26.9%) in the total lipids was high (≥20%) during days 1–3.

The effect of initial glucose concentration was evaluated in five day batch cultures (Appendix A). An increase in the initial glucose concentration from 20 to 30 g/L had no effect on biomass concentration, but it increased the total lipid content of the biomass by 1.28-fold while the total carotenoid content decreased by 15%. Further increases in initial glucose concentration (40–50 g/L) had no significant effect (*p* > 0.05) on biomass concentration and its content of lipids and carotenoids, as glucose was not fully consumed by day five (Appendix A).

### 2.3. Thraustochytrium sp. RT2316-16 Genome Sequencing

The sequencing resulted in 21,556,136 reads, ranging in length from 35 to 151 base pairs. The trimming operation removed 817,253 reads (3.8%). The assembly resulted in 9532 scaffolds with an N50 value of approximately 17 kbp. The largest assembled contig had a length of 344 kbp. This, combined with the high N50 value, indicated a low fragmentation level (Table 3). The estimated genome size of *Thraustochytrium* sp. RT2316-16 was 44 Mb (Table 3).

A search for conserved orthologs genes (BUSCO) showed that around 15.3% of the genes were missing (Table 4) and the low number of duplicated (3%) and fragmented (8.6%) BUSCO genes (Table 4) corroborated the low level of fragmentation observed in the assembly statistics (Table 3).

The annotation procedure resulted in 5656 coding sequences predicted from the genome of *Thraustochytrium* sp. RT2316-16. Of these, 4559 genes showed homology with sequences deposited in public databases (Appendix A). The identification of non-coding RNA genes, found four copies of 18S and 28S rRNA genes, 39 copies of 8S rRNA gene, and 80 copies of transfer RNAs (tRNA) in the genome. The distribution of allele frequencies strongly suggests that the genome of *Thraustochytrium* sp. RT2316-16 is haploid.

The analysis of coding sequences with KEGG mapping tools found 1284 enzymes distributed among 12 metabolisms (Appendix A). In carbohydrate metabolism, 40.7% of the enzymes related to “Glycolysis/Gluconeogenesis”, “Propanoate metabolism”, “Pyruvate metabolism”, and “Inositol phosphate metabolism” (Appendix A). Around 38.3% of the enzymes related to amino acids metabolism (Appendix A), including: 33 enzymes associated with “Glycine, serine and threonine metabolism”; 31 enzymes associated with “Cysteine and methionine metabolism”; and 36 enzymes linked with “Valine, leucine and isoleucine degradation”. Enzymes relating to lipid metabolism were mainly concerned with “Fatty acid degradation” (16.5%) and “Glycerophospholipid metabolism” (17.7%) (Appendix A).

### 2.4. Biosynthesis of Fatty Acids in Thraustochytrium sp. RT2316-16

Annotated genes involved in the biosynthesis of saturated fatty acids (palmitic acid, PA, C16:0; stearic acid, SA, C18:0) in RT2316-16 are shown in Appendix A. Most of the genes coding for enzymes are involved in the elongation and desaturation pathway for synthesis of long chain ω-3 PUFA, starting from palmitic acid, were found in the genome of RT2316-16. Δ5-Desaturase is a key enzyme for synthesis of EPA and DHA via the elongation and desaturation pathway. The gene Thraus_T4048, identified as coding for an acyl-lipid (8-3)-desaturase (D5FAD_THRSP; Appendix A, was translated into protein and queried by homology against non-redundant protein database in NCBI using the BLASTP algorithm (https://blast.ncbi.nlm.nih.gov, accessed on 12 December 2020). Results showed a high identity match (63.8%) with a Δ5-desaturase of *Thraustochytrium aureum* (accession BAK08911.1).

A homology search against a non-redundant protein database in NCBI using BLASTP algorithm (https://blast.ncbi.nlm.nih.gov, accessed on 1 January 2021) was performed in the genome of RT2316-16 using the sequence of a PUFA synthase subunit A from *Thraustochytrium* sp. ATCC 26185 (UniProt A0A1B3PEI6) as a query. This resulted in a significant hit with Thraus_T4284, which was annotated as putative inactive phenolphthiocerol synthesis polyketide synthase type I PKS, a function that was confirmed by the search of conserved domains in NCBI. Four significant domains were found: PKS (2.58 × 10^−147^), ketoreductase (1.16 × 10^−44^), a/b hydrolase (1.74 × 10^−30^), and phosphopantetheine binding (3.98 × 10^−6^).

### 2.5. Carotenoid Biosynthesis by Thraustochytrium sp. RT2316-16

All genes coding for the mevalonate pathway enzymes, involved in the synthesis of geranylgenranyl diphosphate from acetyl-CoA, were found in the genome of RT2316-16 (Appendix A). The gene Thraus_T3283, identified as a carotenoid 3,4-desaturase (CRTD_HALJT, Appendix A), was translated into protein and queried by homology against a non-redundant protein database in NCBI using the BLASTP algorithm. Results showed a high identity match (59%) and similarity (72%) to the β-carotene synthase of *Aurantiochytrium* sp. KH105 (accession BBB35234.1). This enzyme is coded by *crtIBY*, a gene that has been reported also in the genomes of other thraustochytrids [24].

A pairwise alignment between the Thraus_T3283 and *crtIBY* genes in *Aurantiochytrium* sp. KH105 (accession BBB35234.1) is shown Appendix A. Two conserved domains could be identified. The first spanned the amino acid residues 53 to 558 which was identified as a ctrI superfamily (4.13 × 10^−92^), a bacterial-type phytoene desaturase that converts phytoene to lycopene. The second domain spanned the residues 621 to 780, and was identified as a member of the isoprenoid biosynthesis superfamily (1.63 × 10^−27^). This region was also identified as a squalene/phytoene synthase in *Aurantiochytrium* sp. KH105. In Thraus_T3283, the two lycopene cyclase domains, observed in the *Aurantiochytrium* sp. KH105 sequence, were missing. Genes coding for the enzymes involved in the synthesis of astaxanthin from β-carotene (i.e., β-carotene ketolase, EC 1.14.99.63; β-carotene hydrolase, EC 1.14.15.24) were not found in RT2316-16.

## 3. Discussion

The final biomass concentrations of the different pale strains were significantly different (*p* < 0.05), ranging between 1.3 ± 0.2 (RT2316-22) and 3.8 ± 0.1 g/L (RT2316-24) (Table 1). The low lipid content of the biomass (<20% *w/w*) of the pale isolates suggested they were not oleaginous, although the growth environment used (e.g., medium composition, incubation period) may have contributed to the observed low lipid level. The pale isolates differed significantly (*p* < 0.05) in percentages of EPA and DHA within the total lipids. The isolate RT2316-26, with a 98.8% identity match to *Oblongichytrium* sp. (Table 1), had the highest percentage of DHA (39.2 ± 2.8%) and EPA (18.2 ± 1.0%) in the total lipid (Table 1). Four of the carotenogenic isolates accumulated lipids at >20% *w/w* in the biomass (Table 2) and the highest lipid content (26.3 ± 1.7% *w/w*) occurred in the biomass of RT2316-16. On average, total lipids of the carotenogenic strains (Table 2) contained less EPA and DHA than lipids from the pale strains (Table 1). However, there were two exceptions, namely carotenogenic isolates RT2316-38 and RT2316-40; RT2316-38 had 50.2 ± 2.7% DHA in total lipids, whereas RT2316-40 had 47.5 ± 0.0%. Unfortunately, both isolates grew poorly, achieving a final biomass concentration of less than 2 g/L under the conditions used.

The combination of a relatively high biomass concentration (7.2 g/L; Table 2), a high lipid content of the biomass (26.3%; Table 2), and a high total carotenoid level in the biomass (63.9 μg/g) of the strain RT2316-16 were encouraging, leading to its selection as the focus of this study. For otherwise identical conditions, the strain RT2316-16 had a total carotenoid productivity that was ~37% higher than the next best strain (i.e., strain RT2316-49; Table 2). In carotenogenic thraustochytrids, the total carotenoid content in the biomass has been reported to range from 5.7 μg/g to 450 μg/g [25]. In the best reported case, the *Thraustochytrium* sp. CHN-1 grown at 23 °C under light accumulated a peak total carotenoid level of ~450 μg/g on day 8 of a batch culture, but the biomass concentration was only 1.5 g/L [14]. The carotenoid content in the biomass declined if the culture was prolonged beyond 8 days [14]. The observed peak carotenoid level in *Thraustochytrium* sp. CHN-1 translated to a maximum carotenoid productivity of around 84 μg/(L day). In comparison with this, the total carotenoid productivity of the strain RT2316-16 (Table 2) was 92 μg/(L day). Therefore, the strain RT2316-16 could be concluded to be an excellent producer of carotenoid with a strong potential for further productivity enhancements through culture optimization.

The fatty acid composition of the total lipids in RT2316-16 changed during cultivation. The content of EPA+DHA and palmitic acid followed opposite trends during growth. The percentage of palmitic acid increased during the growth phase and decreased after nitrogen exhaustion. Preferential usage of saturated fatty acids after the exhaustion of glucose (Figure 1b) may have contributed to the observed late increase in the proportion of the long chain ω-3 PUFA in the total lipids (Figure 2). A late increase of PUFA content was reported also in *Schizochytrium* sp. S31 [8]. In this thraustochytrid, saturated fatty acids in neutral lipids were preferentially consumed, once the carbon supply in the culture medium was exhausted, to generate PUFA-rich phospholipids [8]. As phospholipids are main components of cell membranes, this response would allow production of new cells, or cells better adapted to survive famine.

In an earlier report, astaxanthin in the biomass of *Thraustochytriidae* sp. AS4-A1 (similar to *Ulkenia*) was found to decrease if the composition of the medium was modified to increase biomass growth by adding nitrogen sources such as yeast extract and monosodium glutamate [16]. Similarly, in *Schizochytrium* KH105, a low nitrogen concentration, presumably paralleled with slowed growth, was found to promote astaxanthin accumulation [13]. Synthesis of carotenoids requires acetyl-CoA (to produce farnesyl diphosphate; Appendix A) that is available when its rate of production exceeds the consumption for building biomass precursors needed during growth. This suggests that the accumulation of carotenoids should be favored if growth is suppressed by nutrient limitation in such a way that acetyl-CoA continues to be made. However, this was not the case in RT2316-16. An increase in initial concentration of only the glucose (from 20 to 30 g/L) increased the total lipid content of the biomass by 1.28-fold, but the total carotenoid content decreased by 15% (Appendix A). Thus, a copious supply of the carbon source in combination with nitrogen limitation resulted in more of the acetyl-CoA being channeled into lipids instead of carotenoids. These preliminary results suggest that the total lipid in the biomass of RT2316-16 may be increased in a glucose fed culture that maintains the glucose concentration at a high level (above 10 g/L). In contrast, if both glucose and nitrogen are fed, the biomass concentration should increase together with the total production of carotenoids.

The estimated genome size of *Thraustochytrium* sp. RT2316-16 (Table 3) was comparable to that of other thraustochytrids: *Aurantiochytrium* sp. T66, 43 Mb [26]; thraustochytrid strain CCAP_4062/3, 38.7 Mb [27]; *Aurantiochytrium* sp. SK4, 49.62 Mb [28]; *Schizochytrium* sp. CCTCC M209059, 39.09 Mb [29]; *Schizochytrium* sp. S31, 42.99 Mb [8]; and *Thraustochytrium* sp. ATCC 26185, 39 Mb [30]. However, it was smaller than the genome of *Schizochytrium* sp. (Mn4), 65.69 Mb; *Thraustochytriidae* sp. (SW8), 61.67 Mb [31]; and *Aurantiochytrium* sp. KH105, 95 Mb, a possible diploid strain [24].

In some thraustochytrids (*Schizochytrium* sp. [32]; *Thraustochytrium* sp. SZU445 [33]), the synthesis of long-chain ω-3 PUFA from acetyl-ACP and malonyl-ACP is accomplished by polyketide synthase (PKS) systems, also known as PUFA synthases. Our results implied that the synthesis of long chain ω-3 PUFA in RT2316-16 was not due to a PKS system. Most of the genes coding enzymes in the alternative pathway (elongation and desaturation pathway) were found in the genome of RT2316-16 with two exceptions, the genes coding for Δ15-desaturase and the very-long-chain enoyl-CoA reductase. The enzyme Δ15-desaturase, a ω-3-desaturase, catalyzes desaturation of linoleic acid (LA, C18:2n6) producing α-linolenic (ALA, C18:3n−3) precursor of the ω-3 fatty acid cascade. The very-long-chain enoyl-CoA reductase catalyzes the last of the four reactions in the elongation cycle of very long-chain fatty acids (C > 18). In this cycle, the condensing reaction is the rate-limiting step that is catalyzed by enzymes which act on specific fatty acyl-CoA (the substrate) [34]. In contrast, the other three enzymes (i.e., 3-ketoacyl-CoA reductase, 3-hydroxyacyl-CoA dehydratase, and trans-2-enoyl-CoA reductase or very-long-chain enoyl-CoA reductase) that catalyze the reduction, dehydration and final reduction reactions have broad substrate specificities [35]. If the long-chain ω-3 PUFA in RT2316-16 are synthesized through the elongation and desaturation pathway, then the last reduction step in the elongation cycle would require an unidentified enzyme.

In *crtIBY*, the gene coding for β-carotene synthase of *Aurantiochytrium* sp. KH105 [24], the following three genes are fused: *crtB* (coding for geranylgeranyl phytoene synthase), *crtI* (coding for phytoene desaturase), and *crtY* (coding for lycopene cyclase). This gene organization likely improves the efficiency of β-carotene synthesis since the intermediates, such as lycopene, are not the target metabolites [24].

Unlike *Aurantiochytrium* sp. KH105, an astaxanthin producer, RT2316-16 is a producer of β-carotene and canthaxanthin. In certain microalgae synthesis of ketocarotenoids is due to substrate preferences of β-carotene ketolase. For example, the enzyme of *Haematococcus pluvialis* produces canthaxanthin via echinone, while the diketolase of *Chlamydomonas reinhardtii* is able to convert β-carotene into canthaxanthin and zeaxanthin into astaxanthin [36]. In the genome of RT2316-16, genes coding for ketolases similar to those reported in microalgae were not found, suggesting that different enzymes were involved in the synthesis of the ketocarotenoid. In the yeast *Xanthophyllomyces dendrorhous* (*Phaffia rhodozyma*), a single oxygenase (called astaxanthin synthase) converts β-carotene into astaxanthin [37]. This enzyme is a cytochrome P450 belonging to the cytochrome P450 3A subfamily. In RT2316-16, the gene coding for a cytochrome P450 3A14 (Thraus_T3181) (Appendix A) may have had a role in the synthesis of canthaxantin.

## 4. Materials and Methods

### 4.1. Collection and Isolation of Thraustochytrids

Seawater samples were collected in sterile bottles from coastal areas near the following five Antarctic bases: (1) Professor Julio Escudero (BE; S 62°12′57″ E 58°57′35″); (2) Gabriel González Videla (BGV; S 64°49′26.70″ W 62°51′28.40″); (3) Yelcho (BY; S 64°52′33.18” W 63°35′3.4”); (4) Prat (BP; S 62°28′44.3″ W 59°39′53.6″); and (5) O’Higgins (BO; S 63°19′12.2″ W 58°57′45.5″). Additional other sampling locations were the coastal waters of Livingstone Island (LI; S 62°38′48.72″ W 60°22′35.07″), Deception Island (DI; S 62°58′42″ W 60°33′36″), and Barrientos Island (BI; S 62°24′30″ W 59°45′18″). All locations were visited during the Antarctic Scientific Expedition 54 in February 2018. The samples were collected at depths of between 0.3 and 1 m. The seawater temperature at sampling locations ranged from 0.5 to 2 °C.

Samples supplemented with sterile pine pollen (~75 mg/L) were incubated at 5 °C for 15 days. Afterwards, the pollen grains with attached microorganisms were recovered by filtration (0.45 μm nominal pore size) and dispersed in a solid medium. This medium comprised of the following components (g/L): glucose (Merck, Darmstadt, Germany) 1, yeast extract (BBL™, Becton, Dickinson and Co., NJ, USA) 6, and agar (Merck) 15, in artificial seawater (ASW [38]) diluted to 50% *v/v* with distilled water. Streptomycin sulfate and penicillin G (Sigma, St. Louis, MO, USA) (0.3 g/L, each) were added to prevent proliferation of the bacteria. Agar plates were held at 20 °C until visible colonies appeared. The incubation temperature was relatively high compared to the temperature of the natural habitat (0.5–2.0 °C) as the focus was on recovering strains that would grow rapidly under commercially practicable conditions. Individual colonies were harvested from agar and cultured on fresh plates of the above specified medium until pure isolates (by microscopic inspection) were obtained. The isolated microorganisms were grown at 20 °C in sterile test tubes containing 5 mL of a liquid medium (glucose 2 g/L, yeast extract 6 g/L, monosodium glutamate (Merck) 0.6 g/L, in half-strength ASW). Stock cultures, prepared by transferring 0.5 mL of a grown culture in microtubes containing sterile glycerol (Merck) (0.5 mL), were maintained at −80 °C for up to three months.

### 4.2. DNA Extraction and Molecular Identification of Isolates

One milliliter of each stock cultures was used to inoculate an Erlenmeyer flask (250 mL) containing 100 mL of a sterile medium (glucose 20 g/L, yeast extract g/L, monosodium glutamate 0.6 g/L in ASW 50% *v/v*). Trace elements and vitamins were added to the medium with the inoculum as previously specified [38]. The flasks were incubated at 15 °C in an orbital shaker (150 rpm) (Zhicheng, Shanghai, China). DNA was isolated from cells harvested after 7 days using the UltraClean™ Kit (Mo Bio Laboratories Inc., Carlsbad, CA, USA) following the manufacturer’s instructions. The gene sequence that encodes the 18S rRNA was amplified using published PCR conditions [39] with the modifications noted here. Primers used were: forward (F) FA1 (5′-AAAGATTAAGCCATGCATGT-3′), and FA2 (5′-GTCTGGTGCCAGCAGCCGCG-3′), and reverse (R) RA2 (5′-CCCGTGTTGAGTCAAATTAAG-3′) and RA3 (5′-CAATCGGTAGGTGCGACGGGCGG-3′). Two PCR reactions were performed to obtain more specific amplification products. PCR program for FA1-RA2 was: 3 min at 95 °C; 30 cycles of 1 min at 94 °C, 1 min at 61 °C, 1 min at 72 °C; and 10 min at 72 °C. PCR program for FA2-RA3 was: 3 min at 95 °C; 30 cycles of 1 min at 94 °C, 1 min at 69 °C, 1 min at 72 °C; and 10 min at 72 °C. PCR products were purified using the E.Z.N.A Gel Extraction Kit (Omega Bio-tek, Inc., Norcross, GA, USA), following the manufacturer’s instructions and Sanger sequencing was performed by Macrogen Corporation (Rockville, MD, USA). Results were assembled using BioEdit 7.2 software [40] and compared with sequences in EMBL/DDBJ/PDB/GenBank databases by BLASTN 2.2.21 analysis. A phylogenetic tree was generated by phylogeny.fr [41] (http://www.phylogeny.fr, accessed on 31 December 2020), using MUSCLE, ProtDist/FastDist+BioNJ (distance-based method) and TreeDyn, for multiple sequence alignment, tree construction, and tree visualization, respectively.

### 4.3. Genome Sequencing and Assembly

The genome of the isolate codenamed RT2316-16 was sequenced (Genoma Mayor, Santiago, Chile) using a MiSeq platform in a paired-end mode for 150 cycles. The resulting files containing the sequences and their respective base-calling accuracy were processed using TrimGalore (Version No. 0.6.5, Babraham Institute Enterprise (BIE), https://www.bioinformatics.babraham.ac.uk/projects/trim_galore/, accessed on 1 March 2020) [42] to remove all residual adaptor sequences and reads with an average Q-Score less than 30. Before and after the trimming steps, the reads’ overall quality were visualized using FastQC (Version No. 0.11.9, https://www.bioinformatics.babraham.ac.uk/projects/fastqc/) (accessed on 1 March 2020) [43]. Then the reads that passed the quality control step were used to reconstruct the genome using the Abyss 2.0 software (https://github.com/bcgsc/abyss-2-giab, accessed on 1 March 2020) [44]. Genome’s quality was evaluated using assembly metrics by QUAST (http://quast.sourceforge.net/quast, accessed on 1 March 2020) [45], and its completeness was assessed with BUSCO [46].

### 4.4. Data Availability

The raw data from the high throughput sequencing were deposited in the Sequence Read Archive (SRA) of NCBI under the BioProject accession number PRJNA719725.

### 4.5. Gene Prediction and Annotation

Contigs resulting from the assembly steps were annotated using MAKER2 [47]. An initial step using EXONERATE [48] providing protein evidence from the Thraustochytriales order deposited in UniProtKB was performed. These data were then used to train gene models as input for AUGUSTUS [49], SNAP [50], and glimmerHMM [51] for ab initio predictions. These predictions were evaluated by Evidence Modeller [52] to obtain a consensus of genes predicted for the genome. For the identification of non-coding RNA genes, the ARAGORN algorithm [53] was used to predict transfer RNA (tRNA) genes within the genome. RNAMMER v1.2 [54] was used to identify 8S, 18S and 28S ribosomal RNAs (rRNA).

The ploidy of the genome was calculated aligning the trimmed reads to the assembled genome using bowtie2 [55]. The allele frequencies were calculated and their distribution was analyzed using ploidyNGS [56].

### 4.6. Production of Biomass, Lipids and Carotenoids

The inoculum for the culture experiments was prepared in Erlenmeyer flasks (250 mL) containing 100 mL of a sterile medium (glucose 20 g/L, yeast extract 6 g/L, monosodium glutamate 0.6 g/L, in half-strength ASW). A 1 mL portion of the respective pure stock culture was added. At inoculation, sterile trace minerals and vitamins solutions were added to the culture [38]. The inoculated flask was incubated at 15 °C in an orbital shaker (150 rpm) for five days. The grown culture (5 mL) was used to inoculate fresh sterile medium (100 mL) in 250 mL Erlenmeyer flasks. After seven days of incubation (150 rpm, 15 °C), the biomass was recovered by centrifugation (7000× *g*, 4 °C, 10 min), washed (1 × 10 mL distilled water), recovered by centrifugation, frozen, lyophilized, weighed, and stored at −20 °C for further analysis.

The inoculum for the growth curve of *Thraustochytrium* sp. RT2316-16 was prepared as described above. A 5 mL portion of the grown culture was used to inoculate 20 Erlenmeyer flasks (250 mL each), each containing 100 mL of the above specified sterile medium. The flasks were incubated in an orbital shaker (150 rpm, 15 °C). Two flasks were withdrawn each 24 h, the biomass was recovered by centrifugation, washed with distilled water, lyophilized, weighed, and stored at −20 °C for further analysis. The supernatant was filtered (0.2 μm nominal pore size PTFE membrane) and frozen −20 °C) for further analysis. In separate experiments, the effect of the initial glucose concentration (20, 30, 40 and 50 g/L) on the production of biomass, and lipid and carotenoid contents was assessed after five days of incubation (150 rpm orbital shaker, 15 °C).

### 4.7. Concentrations of Biomass, Residual Sugars and Amino Acids

Concentration of biomass (dry weight) was measured gravimetrically by recovering the cells by centrifugation (7000× *g*, 4 °C, 10 min) from a known volume of culture (ca. 10 mL). The pellet was washed twice by resuspending in distilled water (5 mL per wash), recovered by centrifugation, and dried to constant weight at 65 °C. Concentration of residual glucose was measured by HPLC analysis (Alliance Waters e2695 Separation Module; Waters Inc, Milford, MA, USA) of the culture supernatant using a Sugar HPX-87H column held at 65 °C. The mobile phase was sulfuric acid (5 mM) at a flow rate of 0.6 mL/min. A refractive index (Waters Inc., Milford, MA, USA) detector was used and glucose standard solutions were employed for calibration.

Total amino acids concentration in the culture supernatant was determined spectrophotometrically. The supernatant sample (100 µL) was mixed with 1 mL of o-phthalaldehyde (OPA; Sigma) reagent (5 mg OPA dissolved in 100 µL of pure ethanol, 5 µL of β-2-mercaptoethanol and 10 mL of 50 mM carbonate buffer, pH 10.5) and the absorbance was measured at 340 nm exactly 2 min after mixing [57]. The blank was made as above, but with the sample replaced by half-strength ASW. Standard solutions of L-lysine (Sigma) in diluted ASW were used to make the calibration curve.

### 4.8. Extraction of Total Lipids and Fatty Acid Profile Determination

The total lipids in the biomass was extracted using the Bligh and Dyer method [58]. A 50 mg portion of the freeze-dried biomass was extracted (1 h, 150 rpm) with 9.5 mL of a solvent mixture of chloroform:methanol:phosphate buffer (50 mM, pH 7.4) 2.5: 5.0: 2.0 by volume.

This slurry was transferred to a separating funnel containing 2.5 mL of chloroform and mixed. A 2.5 mL portion of the phosphate buffer was then added and mixed. The chloroform layer was recovered and evaporated at room temperature to obtain the mass of the dissolved lipid.

The extracted lipids were methylated using a mixture of alkaline methanol (2 M KOH in methanol) and extracted into petroleum ether. The ratio of alkaline methanol to petroleum ether was 1:10 by volume. The methylation-extraction system was thoroughly mixed (2 min) at room temperature and allowed to stand for 1 h. The petroleum ether layer was recovered by centrifugation (10,000× *g*, 4 °C, 5 min) and evaporated at room temperature in a fume hood to recover the fatty acid methyl esters (FAME).

The FAME profile was determined using a gas chromatographer (GC-2010 Plus; Shimadzu, Kyoto, Japan) equipped with a flame ionization detector and a split injector. A fused silica capillary column (Rtx-2330; 60 m × 0.32 mm × 0.2 μm film thickness; Thames Restek, Saunderton, UK) was used. Nitrogen was the carrier gas. The column temperature profile was as follows: 140 °C for 5 min, then increased to 240 °C at 3 °C/min and held at this temperature for 5 min. The injector and the detector were held at 260 °C and FAME were identified with reference to a 37-component standard FAME Mix (Supelco, Bellfonte, PA, USA). Finally, individual fatty acids were reported as the percentage of the total fatty acids in the total lipids.

### 4.9. Extraction and Quantification of Carotenoids

A culture aliquot (3–5 mL) was centrifuged (2057× *g*, 10 min) and the supernatant was discarded. The cell pellet was mixed with 1 mL of a salt solution (300 mM NaCl in 50 mM phosphate buffer, pH 8.0), vortex mixed for 30 s then sonicated for 20 min (E60H, Elmasonic; Elma Schmidbauer GmbH, Singen, Germany). This suspension was centrifuged to recover the solids. The solids were extracted with 1 mL of a methanol/chloroform mixture (2:1 *v/v*) for 1 h with continuous mixing on a vortex mixer (Velp Scientifica, Usmate Velate, Italy). The supernatant was recovered by centrifugation. Extraction was repeated until the pellet was white. The pooled methanol–chloroform supernatant was stored at 4 °C. Spectrophotometric absorbance of this solution was measured at 460 nm (*A_460_*) against a blank of methanol/chloroform (2: 1 *v/v*) [59]. Total carotenoids (*TC*) in dry biomass (*DW*) was calculated using the following equation [59]:
(1)$$TC\left(\frac{\mu g}{g\;DW}\right)=\frac{{5.405A}_{460}V}{X}$$
where *V* was the volume (mL) of the colored extract, *X* (g) was the quantity of dry biomass extracted in the culture aliquot, and 5.405 was the average molar extinction coefficient of the following ten carotenoids: lycopene, α-carotene, β-carotene, γ-carotene, zeaxanthin, rhodoxanthin, astaxanthin, lutein, α-apo-2-carotenal, and dihydro-α-carotene.

### 4.10. Thin Layer Chromatography (TLC)

The extracted carotenoids from the previous section were saponified and separated on a precoated TLC Aluminiumoxid 60 F254 neutral plate (Merck). For saponification, the dry pigment was recovered by evaporating the extraction solvent (see Section 4.9), and dissolved it in 500 μL of a methanol/chloroform mixture (2: 1 *v/v*). A 50 μL portion of alkaline methanol (112.2 g KOH per L methanol) was added. This mixture was incubated at 40 °C for 30 min. The liquid phase was recovered by centrifugation (10,000× *g*, 10 min) and loaded on the TLC plate. Solutions of authentic astaxanthin, canthaxanthin, and β-carotene (Sigma) were used as standards. The mobile phase was 75:25 by volume mixture of petroleum ether and acetone. Chromatograms were developed at room temperature in a glass chromatographic chamber saturated with the vapor of the mobile phase. The developed plate was dried in the air in the dark.

### 4.11. Statistical Analysis

Variance analysis (ANOVA) followed by a comparison of the means (Ducan’s test) were used to determine significant differences among the isolates at a 5% confidence level.

## 5. Conclusions

The frequency occurrence of carotenogenic thraustochytrids in seawater samples collected at the different locations visited during the Antarctic Scientific Expedition 54 (February 2018) was nearly the same as that of strains that did not accumulate carotenoids. On average, the carotenogenic thraustochytrids produced more biomass and accumulated more lipids, albeit with a lower content of DHA and EPA than the pale strains. Among the isolates, strain RT2316-16 identified as *Thraustochytrium* sp. was the best candidate for producing lipids containing EPA, DHA, and carotenoids (β-carotene and canthaxanthin). Genome annotation suggested that this isolate synthesized long-chain ω-3 PUFA via the elongation and desaturation pathway. As in other thraustochytrids that synthesize astaxanthin, an enzyme closely related to β-carotene synthase was involved in the synthesis of carotenoids in RT2316-16. Different domains in the gene coding for β-carotene synthase in RT2316-16, and the absence of a gene coding for β-carotene hydrolase, putatively explained the synthesis of canthaxanthin instead of astaxanthin. As canthaxanthin is a strong antioxidant and an approved food colorant in some countries, further studies for optimizing its production by the natural producer RT2316-16 are recommended.

## Figures and Tables

**Figure 1 marinedrugs-19-00386-f001:**
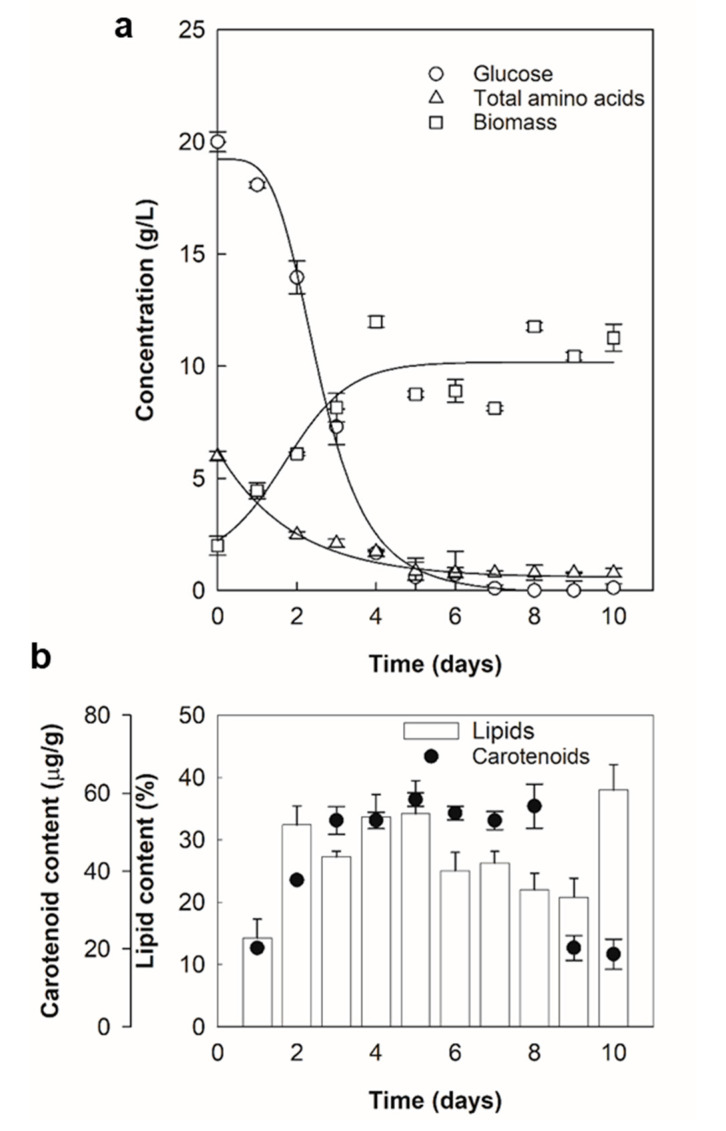
Profiles of concentrations of biomass, residual glucose, and residual amino acids (**a**); and total lipids and carotenoids in the biomass (**b**) of *Thraustochytrium* sp. RT2316-16. Growth temperature was 15 °C.

**Figure 2 marinedrugs-19-00386-f002:**
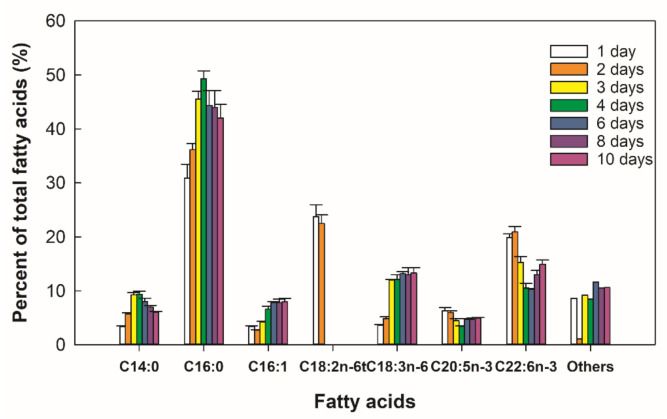
Fatty acid composition (%) of total lipids in the biomass of *Thraustochytrium* sp. RT2316-16 incubated for various periods at 15 °C. C14:0, myristic acid; C16:0, palmitic acid; C16:1, palmitoleic acid; C18:2n-6t, linolelaidic acid (all-trans-Δ^9, 12^); C18:3n-6, γ-linolenic acid (all-cis-Δ^6, 9, 12^); C20:5n-3, eicosapentaenoic acid (all-cis-Δ^5, 8, 11, 14, 17^); and C22:6n-3, docosahexaenoic acid (all-cis-Δ^4, 7, 10, 13, 16, 19^).

**Table 1 marinedrugs-19-00386-t001:** Percent identity match of pale isolates to closest relative in the GenBank, accession numbers of isolates, and culture attributes.

Strain	Closest Relative in GenBank	Identity Match (%)	Accession	*X* (g/L)	*TL* (%)	*EPA* (%)	*DHA* (%)
RT2316-14	*Oblongichytrium* sp.	98.45	MT648446	2.7 ± 0.0 ^b^	16.0 ± 1.3 ^a,b^	9.9 ± 0.3 ^d^	26.1 ± 1.2 ^d^
RT2316-15	*Oblongichytrium* sp.	99.32	MT648448	3.7 ± 0.2 ^a^	16.1 ± 0.5 ^a,b^	8.4 ± 0.2 ^e^	21.6 ± 1.7 ^e^
RT2316-18	*Thraustochytrium* sp.	100.00	MT648499	2.7 ± 0.1 ^b^	12.3 ± 0.3 ^b,c^	7.1 ± 0.5 ^f^	20.8 ± 1.2 ^e^
RT2316-21	*Oblongichytrium* sp.	99.47	MT667280	2.0 ± 0.2 ^c^	9.9 ± 1.0 ^c^	10.4 ± 1.9 ^d^	22.0 ± 1.0 ^e^
RT2316-22	*Oblongichytrium* sp.	98.92	MT667372	1.3 ± 0.2 ^d^	9.3 ± 0.1 ^c^	9.1 ± 0.0 ^d^	22.8 ± 0.0 ^e^
RT2316-23	*Oblongichytrium* sp.	99.46	MT667375	1.8 ± 0.1 ^c^	9.8 ± 1.2 ^c^	13.3 ± 0.1 ^c^	37.4 ± 1.9 ^a^
RT2316-24	*Oblongichytrium* sp.	99.46	MT667352	3.8 ± 0.1 ^a^	13.4 ± 0.2 ^b^	1.3 ± 0.0 ^g^	2.1 ± 0.0 ^f^
RT2316-25	*Oblongichytrium* sp.	99.19	MT667384	2.2 ± 0.2 ^c^	14.1 ± 1.0 ^b^	13.2 ± 0.2 ^c^	29.5 ± 0.9 ^c^
RT2316-26	*Oblongichytrium* sp.	98.81	MT667421	3.9 ± 0.1 ^a^	10.2 ± 0.1 ^c^	18.2 ± 1.0 ^a^	39.2 ± 2.8 ^a^
RT2316-28	*Aurantiochytrium* sp.	98.34	MT668501	1.2 ± 0.1 ^d^	15.9 ± 1.4 ^a,b^	ND	ND
RT2316-29	*Oblongichytrium* sp.	99.78	MT668508	2.9 ± 0.0 ^b^	13.8 ± 0.9 ^b^	8.2 ± 0.0 ^e^	28.8 ± 0.0 ^c^
RT2316-31	*Oblongichytrium* sp.	98.24	MT668540	3.0 ± 0.2 ^b^	17.3 ± 0.8 ^a^	14.8 ± 0.3 ^b^	33.8 ± 2.2 ^b^

*X*, Biomass concentration; *TL*, total lipid content in the dry biomass; EPA, percentage of EPA in fatty acids of total lipids; DHA, percentage of DHA in the fatty acids of the total lipids. All cultures were grown for 5 day at 15 °C. ND, Not detected. Different superscript letters within a column denote significant differences (*p* < 0.05).

**Table 2 marinedrugs-19-00386-t002:** Percent identity match of pigmented isolates to closest relative in the GenBank, accession numbers of isolates and culture attributes.

Strain	Closest Relative in GenBank	Identity Match (%)	Accession	*X* (g/L)	*TL* (%)	*TC* (µg/g)	*EPA* (%)	*DHA* (%)
RT2316-37	*Thraustochytrium* sp.	99.48	MT812689	7.1 ± 0.3 ^b^	21.6 ± 0.8 ^b,c^	39.9 ± 1.3 ^c,d^	1.5 ± 0.0 ^b^	3.3 ± 0.4 ^b,c^
RT2316-38	*Thraustochytrium* sp.	99.25	MT812701	1.9 ± 0.1 ^d^	23.2 ± 1.2 ^b^	27.4 ± 5.3 ^e^	6.7 ± 1.1 ^a^	50.2 ± 2.7 ^a^
RT2316-16	*Thraustochytrium* sp.	100.00	MT648462	7.2 ± 0.2 ^b^	26.3 ± 1.7 ^a^	63.9 ± 3.2 ^a^	6.0 ± 0.0 ^a^	4.2 ± 0.0 ^b^
RT2316-45	*Thraustochytrium* sp.	99.45	MT814239	7.3 ± 0.1 ^b^	18.6 ± 1.2 ^c^	35.2 ± 6.6 ^d^	2.6 ± 1.5 ^b^	5.2 ± 1.5 ^b^
RT2316-44	*Thraustochytrium* sp.	99.26	MT814238	8.3 ± 0.2 ^a^	20.1 ± 1.3 ^b,c^	30.3 ± 2.7 ^d^	1.9 ± 0.0 ^b^	4.3 ± 0.4 ^b^
RT2316-17	*Thraustochytrium* sp.	95.33	MT648465	5.8 ± 0.3 ^c^	18.0 ± 0.3 ^c^	46.1 ± 3.2 ^c^	0.9 ± 0.5 ^b^	2.4 ± 1.2 ^c^
RT2316-42	*Thraustochytrium* sp.	99.37	MT812971	1.1 ± 0.1 ^f^	14.3 ± 2.1 ^d^	54.6 ± 1.7 ^b^	1.5 ± 0.5 ^b^	4.1 ± 0.0 ^b^
RT2316-40	*Thraustochytrium* sp.	99.34	MT812947	1.5 ± 0.1 ^e^	18.1 ± 0.5 ^c^	43.7 ± 1.3 ^c^	7.1 ± 0.0 ^a^	47.5 ± 0.0 ^a^
RT2316-49	*Thraustochytriidae* sp.	99.47	MT814293	8.3 ± 0.1 ^a^	18.5 ± 0.7 ^c^	40.5 ± 2.0 ^c,d^	1.6 ± 0.2 ^b^	4.1 ± 0.4 ^b^
RT2316-50	*Aurantiochytrium* sp.	99.27	MT814296	5.8 ± 0.3 ^c^	19.0 ± 0.2 ^c,d,e^	ND	0.8 ± 0.0 ^b^	2.7 ± 0.4 ^c^

*X*, Biomass concentration; *TL*, total lipid content in the dry biomass; *TC*, total carotenoid content in the dry biomass; *EPA*, percentage of EPA in fatty acids of total lipids; *DHA*, percentage of DHA in the fatty acids of the total lipids. All cultures were grown for 5 day at 15 °C. ND, Not detected. Different superscript letters within a column denote significant differences (*p* < 0.05).

**Table 3 marinedrugs-19-00386-t003:** Summary of assembly quality parameters of *Thraustochytrium* sp. RT2316-16 genome.

Parameter	Value
Number of contigs	9532
Contigs ≥ 1000 bp	5133
Contigs ≥ 5000 bp	2074
Contigs ≥ 10,000 bp	1167
Contigs ≥ 25,000 bp	378
Largest contig	344,669 bp
Total length	44,309,409 bp
N50	17,465
L50	601
Guanine-cytosine content (%)	58.85
N’s per 100 kbp	439.38

**Table 4 marinedrugs-19-00386-t004:** BUSCO results: Number of full length genes (Completed BUSCO) with either one or more copies found in the genome of *Thraustochytrium* sp. RT2316-16, and the number of partially fragmented genes (Fragmented BUSCOs) found ^£^.

Parameter	Value
Complete BUSCOs	194 (76.1%)
Single-copy BUSCOs	191 (74.9%)
Duplicated BUSCOs	3 (1.2%)
Fragmented BUSCOs	22 (8.6%)
Missing BUSCOs	39 (15.3%)
Total BUSCO genes	255 (100%)

^£^ Database used: Eukaryota_odb10 database.

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
