# Peer review of "Antarctic Thraustochytrids as Sources of Carotenoids and High-Value Fatty Acids"

_marinedrugs, 2021, doi:10.3390/md19070386_

Round 1

Reviewer 1 Report

The authors present an interesting manuscript about the genomic characterization of Antarctic thraustochytrids as source of carotenoids and high-value fatty acids. I think the study is interesting since present genomic data from potentially valuable protist species not well-known to the date. However, after reading the manuscript, I have some comments and doubts for the authors.

  1. The abstract needs to be improved. The aim of the study is not clearly exposed in the abstract. The results, conclusions and potential application/interest of it could also be better explained. In fact, the title is much selfexplanatory than the abstract, so I suggest to change it. Specially, after reading the abstract, it seems that the main result of the study is that the production of canthaxanthin in the isolate which was the best producer of biomass is higher than astaxanthin, on the contrary of other species. But, it is difficult for me to understand why this result is so important or should be worthy to highlight it as much. Moreover, could the authors added in the abstract any conclusion about the potential used of those Antarctic thraustochytrids as sources of carotenoids and high-value fatty acids?
  2. Introduction is a bit too short, especially the first paragraph. The health effects of EPA and DHA and carotenoids could be better defined since are largely accepted. Specially the term “metabolically important” in lane 37 is quite vague. Moreover, the relevance of the study is poorly described: Are this the first time that the genome for Antarctic thraustochytrids is reported/analyzed?
  3. Regarding Results and Discussion, there are some relevant results that the authors highlight even in the abstract, such as Line 24: Abstract: “Main carotenoids in RT2316-16 were β-carotene and canthaxanthin”, that are placed directly in supplementary matherial, or some results that are directly reported in the Discussion section (figure 1b, the estimated genome size of of Thraustochytrium sp. RT2316-16, etc), not in results. This is really confusing and I think it should be changed. Maybe it should be a good idea just to merge both sections. If not, discussion and results should be clearly separate as different sections.
  4. Just as curiosity, according to their data, do the autor think that this strain can really be exploited as profiable biomass/carotenoid producer? If it is posible to know, how are those values as compared to other species?
  5. Finally, at the end of the introduction, the authors declare: lines 66-70“The genome of the most prolific strain for lipids rich in long chain ω-3 PUFA and carotenoids was sequenced. Also, the selected strain’s metabolic pathways used in the synthesis of these valuable compounds were elucidated in attempts to identify strategies for a future lipid yield improvement, and for monitoring the expression of critical genes under different culture conditions.” Why the authors are sure that these are the critical genes which determine the production capacity if the the authors did not analyze the genome of other strains less producers in other to have something to compare to? Would the authors expect to obtain similar genome in all the strains regardeless their quality as producers? All the characterization data of this specific strain are interesting but I think their meaning is lower or their conexion with the best properties of the strain are difficult to see without any other strain to have a reference. Have the authors any idea of why this strain is a better producer? Are you planning to do something in this way, maybe in the future?

Author Response

Responses to comments of Reviewer 1

Comment 1. The abstract needs to be improved. The aim of the study is not clearly exposed in the abstract. The results, conclusions and potential application/interest of it could also be better explained. In fact, the title is much selfexplanatory than the abstract, so I suggest to change it.

Specially, after reading the abstract, it seems that the main result of the study is that the production of canthaxanthin in the isolate which was the best producer of biomass is higher than astaxanthin, on the contrary of other species. But, it is difficult for me to understand why this result is so important or should be worthy to highlight it as much. Moreover, could the authors added in the abstract any conclusion about the potential used of those Antarctic thraustochytrids as sources of carotenoids and high-value fatty acids?

Response 1. The abstract has been revised to clarify the importance of EPA, DHA and carotenoids and the interest in microbial producers of these compounds (lines 43-46 in page 1). The rationale for the selection of the strain RT2316-16 as the focus of this study has been clarified (lines 49-51 in page 1).

Comment 2.1. Introduction is a bit too short, especially the first paragraph. The health effects of EPA and DHA and carotenoids could be better defined since are largely accepted. Specially the term “metabolically important” in lane 37 is quite vague.

Response 2.1. The “health effects of EPA and DHA” are already well known and this manuscript is not about the health effects of these compounds, or the carotenoids. Nevertheless, the Introduction has been improved with specific mention of some health benefits of the target compounds (lines 67 page 1 – 5 page 2). The term “metabolically important”, has been deleted.

Comment 2.2. Moreover, the relevance of the study is poorly described: Are this the first time that the genome for Antarctic thraustochytrids is reported/analyzed?

Comment 2.2. The rationale for the present study has been further clarified (lines 33-39 in page 2), as requested. There are literally thousands of different “Antarctic thraustochytrids”. The current study is the first report of the genome and culture of the thraustochytrid Thraustochytrium sp. RT2316-16. An earlier study (cited as reference 19) of an Antarctic thraustochytrid focused on an entirely different microorganism that lacked the carotenoids pathway.

Comment 3. Regarding Results and Discussion, there are some relevant results that the authors highlight even in the abstract, such as Line 24: Abstract: “Main carotenoids in RT2316-16 were β-carotene and canthaxanthin”, that are placed directly in supplementary matherial, or some results that are directly reported in the Discussion section (figure 1b, the estimated genome size of of Thraustochytrium sp. RT2316-16, etc), not in results. This is really confusing and I think it should be changed. Maybe it should be a good idea just to merge both sections. If not, discussion and results should be clearly separate as different sections.

Response 3. Presentation of all results has been relocated to the section entitled “Results” (line 47 in page 2) and the discussion has been separated (line 47 in page 7), as requested.

Comment 4. Just as curiosity, according to their data, do the autor think that this strain can really be exploited as profiable biomass/carotenoid producer? If it is posible to know, how are those values as compared to other species?

Response 4. Whether a “strain can really be exploited” is barely relevant to understanding its genome, metabolism and metabolite production. This proviso notwithstanding, the combination of a relatively high biomass concentration, the high lipid content of the biomass and a high level of total carotenoid in the biomass, was the rational for the selection of strain RT2316-16 for further study. This has been clarified in the revised text (lines 12-17 in page 8). The production of carotenoids and other metabolites by various thraustochytrids has been reviewed by Marchan et al. (2018), cited in the revised text as reference 25. A comparative analysis (lines 17-25 in page 8) suggests that the isolate RT2316-16 is indeed a better carotenoid producer than the best previously reported in the literature.

Comment 5. Finally, at the end of the introduction, the authors declare: lines 66-70“The genome of the most prolific strain for lipids rich in long chain ω-3 PUFA and carotenoids was sequenced. Also, the selected strain’s metabolic pathways used in the synthesis of these valuable compounds were elucidated in attempts to identify strategies for a future lipid yield improvement, and for monitoring the expression of critical genes under different culture conditions.” Why the authors are sure that these are the critical genes which determine the production capacity if the the authors did not analyze the genome of other strains less producers in other to have something to compare to? Would the authors expect to obtain similar genome in all the strains regardeless their quality as producers? All the characterization data of this specific strain are interesting but I think their meaning is lower or their conexion with the best properties of the strain are difficult to see without any other strain to have a reference. Have the authors any idea of why this strain is a better producer? Are you planning to do something in this way, maybe in the future?

Response 5. Production capacity of any metabolite in any organisms is ALWAYS a function of the genes present and their expression. Production capacity is determined by “critical genes”, i.e. those that encode enzymes that catalyze the rate limiting steps of the biosynthetic pathway, and those that ensure a supply of the key metabolic intermediates that may be required in the main biosynthetic pathway. Any producer strain must have the complete set of genes that are needed to code all the enzymes involved in each step of the biosynthetic pathway, although the synthetic pathways may be somewhat different in different organisms. If a strain does not produce a certain metabolite, it either lacks one or more of the critical genes, or the genes are not expressed for some reason. We believe a knowledge of annotated genes is a good starting point for manipulating their expression through selection of culture conditions. If one or more key genes of a biosynthetic pathway are missing in a strain, then no amount of manipulation of culture conditions, the medium, or other conventional ‘strain improvement’ strategy will lead anywhere. Concerning comparison with other strains, the productivity of total carotenoid in the strain RT2316-16 has been contextualized with reference to published data for Thraustochytrid sp. CHN-1 which is known to be rich in carotenoids (lines 17-25 in page 8).

Reviewer 2 Report

The authors present an interesting manuscript with valuable data on potentially useful metabolites from a lesser studied group of microorganisms, the thraustochytrids. The manuscript is well written, the methods are clearly described and adequate, the results are presented in a clear way and the discussion reasonable. I recommend publication with only a few edits.

Line 74: define ‘pale isolates’

Line 204 ff: To better understand the differences in final biomass and accumulation of different metabolites, some basic information on the morphology of the different strains would be helpful. E.g. average cell size and shape, different ratio volume/surface, growth rate.

Line 254 ff: the discussion of the genome is very short. Are the strains haploid or diploid, what is known about non-coding sequences etc?

Line 304: Please add information on water depth and temperature for the sampling.

Line 319: isolates were kept at 20C until visible microbial colonies appeared. This temperature is definitely much higher than that of the arctic water. Isn’t there a risk for a sampling bias, and maybe missing microbes growing at lower temperatures?

Author Response

Responses to comments of Reviewer 2

Comment 1. Line 74: define ‘pale isolates’

Response 1. Pale isolates were those that produced white colonies on agar plates, as clarified (lines 50-51 in page 2) in the revised text.

Comment 2. Line 204 ff: To better understand the differences in final biomass and accumulation of different metabolites, some basic information on the morphology of the different strains would be helpful. E.g. average cell size and shape, different ratio volume/surface, growth rate.

Response 2. The requested information has been provided in Table S1 (Supplemental Material): colony size and texture, color of the colony, presence of motile zoospore) for the different isolates. Also, light microscope images (40×) of four the isolated thraustochytrid strains, including RT2316-16, are provided (Figure S1, Supplemental Material).

Comment 3. Line 254 ff: the discussion of the genome is very short. Are the strains haploid or diploid, what is known about non-coding sequences etc?

Response 3. The requested information (analysis of ploidy and non-coding sequence) has been provided in the revised text (lines 28-32 in page 6). According to the bioinformatic analysis the genome of RT2316-16 is haploid.

Comment 4. Line 304: Please add information on water depth and temperature for the sampling.

Response 4. The requested information has been provided in the revised text (lines 5-7 in page 10).

Comment 5. Line 319: isolates were kept at 20C until visible microbial colonies appeared. This temperature is definitely much higher than that of the arctic water. Isn’t there a risk for a sampling bias, and maybe missing microbes growing at lower temperatures?

Response 5. Incubation temperature was indeed 20 °C. This may have eliminated strains that could not grow at this temperature, as acknowledged in the revised text (lines 16-18 in page 10). The focus was on isolating strains capable of growing rapidly under production conditions. A temperature of 5 Celsius or less (i.e. refrigeration temperature) is not conducive to rapid production of biomass in a commercial environment.

Round 2

Reviewer 1 Report

Thank you very much for the efforts of the authors in answering all my comments. I am generally satisfied with all the changes and I appreciate them. On the other hand, I do not think the capital letters in the response 5 was necessary, I think I could have followed the explanation also in lowercase. Anyway, I totally agree that obviously the presence of a certain gen is the first critical step but, as authors also recognized, the regulation of their expression (by both endogenous and exogenous factors), translation into proteins and proteome regulation are also critical and in some cases will determine if a specie is a good or bad producers or whatever. And for this reason, it is really difficult to be sure that the other strains less producers do not have the same genes, when actually those other strains can produce those compounds, although with worse yield. In any case, the new version of the manuscript is much more clear for me and I understand that to know why the selected strain is a better producer that the others was not the main objective of their study, but the genome characterization of the selected strain.